# T cell immunodominance is dictated by the positively selecting self-peptide

Wan-Lin Lo[1], Benjamin D Solomon[2], David L Donermeyer[1], Chyi-Song Hsieh[2], Paul M Allen[1]*

[1]Department of Immunology and Pathology, Washington University School of Medicine, St. Louis, United States; [2]Department of Internal Medicine, Division of Rheumatology, Washington University School of Medicine, St. Louis, United States

**Abstract** Naive T cell precursor frequency determines the magnitude of immunodominance. While a broad T cell repertoire requires diverse positively selecting self-peptides, how a single positively selecting ligand influences naive T cell precursor frequency remains undefined. We generated a transgenic mouse expressing a naturally occurring self-peptide, gp250, that positively selects an MCC-specific TCR, AND, as the only MHC class II I-E$^k$ ligand to study the MCC highly organized immunodominance hierarchy. The single gp250/I-E$^k$ ligand greatly enhanced MCC-tetramer$^+$ CD4$^+$ T cells, and skewed MCC-tetramer$^+$ population toward V11α$^+$Vβ3$^+$, a major TCR pair in MCC-specific immunodominance. The gp250-selected V11α$^+$Vβ3$^+$ CD4$^+$ T cells had a significantly increased frequency of conserved MCC-preferred CDR3 features. Our studies establish a direct and causal relationship between a selecting self-peptide and the specificity of the selected TCRs. Thus, an immunodominant T cell response can be due to a dominant positively selecting self-peptide.

*For correspondence: pallen@wustl.edu

Competing interests: The authors declare that no competing interests exist.

## Introduction

An individual pathogen encodes thousands of potentially immunogenic epitopes, yet during the course of an infection, generally only a fraction of epitopes actually stimulate T cell response (*Sercarz et al., 1993*; *Sant et al., 2007*; *Weaver and Sant, 2009*; *Jenkins et al., 2010*). This fundamental feature of T cell responses is known as immunodominance (*Sercarz et al., 1993*; *Chen et al., 2001*; *Yewdell and Del Val, 2004*; *Yewdell, 2006*; *Sant et al., 2007*; *Weaver and Sant, 2009*; *Jenkins and Moon, 2012*). Most of the effort in immunodominance studies has focused on the antigen presentation and processing pathways (*Deng et al., 1993*; *van Ham et al., 1996*; *Nanda and Sant, 2000*; *Chen et al., 2001*; *Crowe et al., 2003*; *Chen and McCluskey, 2006*; *Lazarski et al., 2006*; *Yewdell, 2006*; *Weaver and Sant, 2009*). Yet immunodominance is also defined by the frequencies of naive T cells specific for individual antigens (*Kotturi et al., 2008*; *Jenkins and Moon, 2012*; *Kwok et al., 2012*). Given that thymic positive selection serves as a major developmental process to generate mature T cell repertoires (*Jameson et al., 1995*; *Morris and Allen, 2012*), one may reason positive selection would play a role in determining naive T cell precursor frequency to dictate immunodominance. However, this possibility has been difficult to address directly, due to several technological issues. First, the frequency of T cells specific for individual peptide–MHC ligands is at most about 20–100 cells per million naive T cells (*Jenkins and Moon, 2012*), making it difficult to identify such an infrequent population. Second, to directly test the potential role of positive selection in immunodominance requires the knowledge of what T cell population specific for a defined antigen a positively selecting ligand is capable of selecting. Therefore, the low frequency of naive T cell precursor and the ignorance of essential information have limited the examination of the roles of positive selecting ligands in dictating naive T cell precursor frequency and immunodominance.

**eLife digest** The immune system protects against disease by recognizing invading pathogens, such as bacteria and viruses, and launching various responses to eliminate them. In vertebrates, such as mice and humans, this response often involves both an 'innate' and an 'adaptive' component. The innate immune response is fast but it is not targeted at the specific pathogen that needs to be eliminated. The adaptive immune response is slower, but it is also stronger and tailored to combat the specific pathogen that is attacking the host. In general, both the innate and adaptive immune responses work together to combat the infection.

The innate immune system is activated when host cells are damaged or stressed, or when 'foreign' molecules that are tell-tale signs of microbes and pathogens are detected. It enlists white blood cells that engulf and digest infected host cells or pathogens, and then display the digested fragments via proteins called major histocompatibility complexes. The adaptive immune response relies on other white blood cells called B cells (which make antibodies) and T cells (which carry out a variety of roles).

Each T cell will respond to a very specific fragment of protein displayed by a major histocompatibility complex. Although a bacteria or virus can generate thousands of protein fragments, only a select few will stimulate an immune response. This phenomenon is known as immunodominance.

As T cells mature in the thymus—a specialized organ located in the chest—they undergo both positive and negative selection. To survive positive selection, the T cells must be able to recognize fragments from the host's own proteins. Negative selection removes T cells that interact too strongly with these 'self-peptides' because such interactions will cause the T cells to destroy the host's tissues and cause auto-immune diseases. Positive selection is thought to have an important role in immunodominance because it shapes the population of T cells released from the thymus, but it has been difficult to test this hypothesis.

Now Lo et al. have explored this question by engineering transgenic mice in which gp250—a self peptide that positively selects T cells that respond to a foreign protein called MCC—was the only self peptide to be presented during positive selection. Subsequent experiments showed that the mature T cells released by the thymus increased the fraction of T cells that were capable of responding to the dominant fraction of possible MCC fragments. By establishing a clear connection between positive selection and immunodominance, the work of Lo et al. could assist the development of more effective vaccines.

The murine CD4$^+$ T cell response against cytochrome $c$ is an outstanding model system to investigate the impact of positively selecting ligands on immunodominance (*Solinger et al., 1979*; *Schwartz, 1985*; *Engel and Hedrick, 1988*). Only one single dominant epitope emerges from the immunization of H−2$^k$ mice with the whole protein of moth cytochrome $c$ (MCC) or closely related pigeon cytochrome $c$ (PCC) (*Hedrick et al., 1982*; *Winoto et al., 1986*; *Hedrick et al., 1988*; *McHeyzer-Williams and Davis, 1995*). Also, the MCC- or PCC-stimulated CD4$^+$ T cell response shows highly organized immunodominance hierarchies. The MCC- or PCC-specific responses are highly dominated by Vα11$^+$ TCR, and exhibit several conserved CDR3 features (*Winoto et al., 1986*; *Hedrick et al., 1988*; *McHeyzer-Williams and Davis, 1995*; *McHeyzer-Williams et al., 1999*; *Mikszta et al., 1999*; *Newell et al., 2011*). During MCC-specific responses, the Vα11$^+$Vβ3$^+$ CD4$^+$ T cells are the most dominant responders, while Vα11$^+$ TCRs pairing with Vβ6$^+$, Vβ8$^+$, or Vβ14$^+$ are the subdominant responders (*Miyazaki et al., 1996*; *Malherbe et al., 2004*). Based on the structural data, certain positions at CDR3α and CDR3β regions, where TCR make contact with MCC peptide, present highly conserved amino acid usages (*McHeyzer-Williams et al., 1999*; *Newell et al., 2011*). These features constitute the strength of utilizing cytochrome $c$ as a model antigen to study CD4$^+$ immunodominance. Moreover, the MCC/I-E$^k$ tetramers have been shown to be able to detect most primary MCC-specific T cells (*Savage et al., 1999*). Importantly, our laboratory had previously identified a naturally occurring positively selecting self-peptide, termed gp250, for its ability to positively select the MCC-specific TCR: AND (*Lo et al., 2009*). In this study, we have generated a transgenic mouse line, the gp250 single chain (SC) mouse, in which the gp250/I-E$^k$ was the only MHC class II ligand presented. Combining MCC

tetramer analysis and our gp250 SC mice permitted us to elucidate the relationship between positively selecting ligands and antigen specificities of post-selection CD4[+] T cell repertoires.

Several studies have attempted to investigate the antigen specificities of the post-selection T cell repertoire by limiting the diversity of positively selecting self-peptides (*Kouskoff et al., 1993*; *Ignatowicz et al., 1996*; *Miyazaki et al., 1996*; *Fukui et al., 1997*; *Grubin et al., 1997*; *Ignatowicz et al., 1997*; *Nakano et al., 1997*; *Surh et al., 1997*; *Tourne et al., 1997*; *Gapin et al., 1998*; *Barton and Rudensky, 1999*; *Chmielowski et al., 2000*; *Barton et al., 2002*; *Huseby et al., 2005*). Studies that limit the diversity of positively selecting self-peptides to a single peptide have involved the introduction of a transgene that encoded a defined peptide covalently linked to MHC class II (*Ignatowicz et al., 1996*, *1997*; *Liu et al., 1997*; *Huseby et al., 2005*), disruption of the peptide exchange molecules H-2M (*Miyazaki et al., 1996*; *Grubin et al., 1997*; *Surh et al., 1997*; *Tourne et al., 1997*), expression of a human invariant chain transgene in which CLIP peptide was replaced with other self-peptides (*Barton and Rudensky, 1999*; *Barton et al., 2002*), or viral expression of altered peptide ligands in the thymus (*Kouskoff et al., 1993*; *Nakano et al., 1997*). Altogether these studies concluded that a single peptide could select a large repertoire of T cells and that the recognition of positively selecting ligands is the driving force behind determining the antigen specificities of post-selection T cell repertoire (*Ignatowicz et al., 1996*; *Fukui et al., 1997*; *Grubin et al., 1997*; *Ignatowicz et al., 1997*; *Surh et al., 1997*; *Fukui et al., 1998*; *Gapin et al., 1998*; *Barton and Rudensky, 1999*; *Chmielowski et al., 2000*; *Barton et al., 2002*; *Huseby et al., 2005*). However, these studies were unable to examine immunodominance because they did not utilize a naturally occurring positively selecting ligand for a defined foreign antigen.

The selection of Vα11[+]Vβ3[+] TCRs was greatly enhanced in our gp250 SC mice. CDR3 sequencing revealed that gp250 skewed the positive selection toward MCC-reactive conserved CDR3 features, especially those CDR3s with serine at α91 and asparagine at β97. Our hypothesis that gp250 favors the positive selection of MCC-reactive T cells was further supported by the greatly expanded MCC-tetramer[+] population in gp250 SC mice. Our data provide direct evidence that positive selection plays a central role in determining the post-selection T cell repertoire and is a major determinant in immunodominance.

## Results

### Generation of a mouse line that expresses gp250 as the only MHC class II ligand

To explore the relationship between positive selection and CD4[+] T cell immunodominance, we utilized the MCC model antigen system. The MCC$_{88-103}$ peptide (referred to MCC hereafter) is an immunodominant peptide following immunization with the MCC protein, and the MCC primary responders and memory cells are exclusively dominated by Vα11[+] T cells (*Winoto et al., 1986*; *Hedrick et al., 1988*; *McHeyzer-Williams and Davis, 1995*; *McHeyzer-Williams et al., 1999*; *Mikszta et al., 1999*; *Malherbe et al., 2004*; *Newell et al., 2011*). The basis for the Vα11[+] dominant response has not been established. The AND TCR is one of the dominant MCC-responsive TCRs and consists of Vα11 and Vβ3 gene segments (*Hedrick et al., 1988*; *Kaye et al., 1989*; *McHeyzer-Williams and Davis, 1995*; *McHeyzer-Williams et al., 1999*; *Mikszta et al., 1999*). Essential for these studies was our previous identification of the naturally occurring positively selecting ligand, gp250, for AND T cells (*Lo et al., 2009*). To examine gp250's effect on the post-selection MCC-specific T cell repertoire, we constructed a transgenic mouse line using the MHC class II promoter in which all of the MHC class II molecules contained the gp250 self-peptide via a covalent linkage of the peptide to the MHC class II β chain. We obtained one B6 founder in which the gp250 covalently linked I-E[k] β chain had successfully integrated and was expressed. The I-E[k] α chain was introduced into the founder by crossing to previously generated I-E[k]α transgenic mice (*Griffiths et al., 1994*). The mice were subsequently bred onto B6 MHC class II[−/−] to eliminate other class II molecules and CD74[−/−] background to assure the gp250/I-E[k] is the only MHC class II ligand (*Ignatowicz et al., 1996*, *1997*). The following experiments were all executed with the transgenic mice on the B6 MHC class II[−/−] and CD74[−/−] background (termed gp250 SC mice hereafter).

In gp250 SC mice, the transgenic covalently linked gp250/I-E[k]β successfully paired with transgenic I-E[k]α, restoring the surface expression of I-E[k] in splenic B cells and CD11c[+] dendritic cells (*Figure 1A–D*). The level of expression of the gp250/I-E[k] was readily detectable, but reconstituted to approximately a 10%

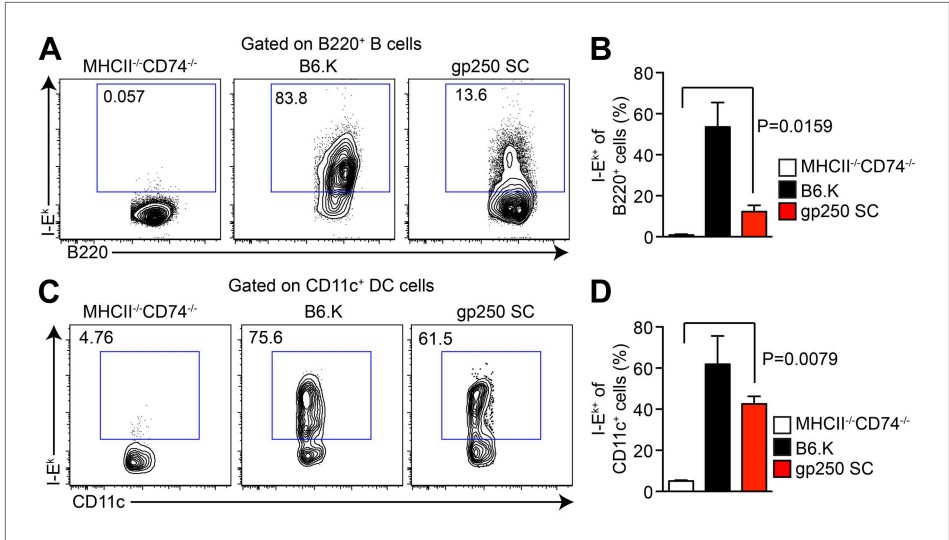

**Figure 1**. The I-E^kα and covalently-linked gp250/I-E^kβ transgenes restore surface expression of I-E^k in gp250 SC mice class II-deficient mice. The surface expression of I-E^k on B cells (**A** and **B**) and dendritic cells (**C** and **D**) from the spleens of gp250 SC mice. The B cells and dendritic cells from MHCII^−/−CD74^−/− mice were used as negative controls. The plots are representative of three independent experiments (**A** and **C**) and summarized in the bar graph (**B** and **D**; mean ± SD; n = 3; two-tailed Mann–Whitney test).

level on B cells and 40% on dendritic cells (*Figure 1A–D*). Immunofluorescence staining of thymic frozen sections of gp250 SC mice showed surface expression of I-E^k in the cortex and medulla of the thymus (data not shown). This level of expression was advantageous for our study in that we had detectable expression, but not supra-physiological levels of a single peptide-MHC and provided us with a suitable window to examine the positive selection mediated by gp250 self-peptide.

The covalently linked gp250/I-E^k positively selected a large number of CD4^+ T cells in the thymus with an average of $1.6 \times 10^8$ total thymocytes and $6.8 \times 10^7$ CD4SP T cells (*Figure 2A–C*). The positive selection efficiency for CD4SP T cells in the gp250 SC mouse was approximately 40% of that seen in the B6.K wild type (*Figure 2B,C*), in line with other published single chain mice. The size of CD8SP T cells in gp250 SC mouse was comparable to that in B6.K (*Figure 2D*), as anticipated given that the gp250 SC mice expressed normal class I molecules. The higher percentage of CD8SP T cells is a reflection of the less efficient CD4SP selection (*Figure 2B–D*).

In the spleen, the peripheral CD4^+ T cell population in gp250 SC mice was readily detectable, being around 40% of the B6.K, yet still averaged $1.7 \times 10^6$ CD4^+ T cells per spleen (*Figure 2E–G*). The single gp250/I-E^k ligand selected a full TCR Vβ repertoire, in that we did not observe any complete absence of Vβs (*Figure 2—figure supplement 1*). However, the frequencies of some TCR Vβs were decreased, including Vβ2^+, Vβ4^+, Vβ8.1/8.2^+, suggesting gp250/I-E^k did not positively select individual Vβs with a uniform selection capability (*Figure 2—figure supplement 1A*). The usages of Vβ5^+, Vβ11^+, and Vβ12^+ were also increased in peripheral CD4^+ T cells in gp250 SC mice (*Figure 2—figure supplement 1B*). Given gp250/I-E^k was the only ligand presented by MHC class II molecules in gp250 SC mice, it was possible that the increased Vβs were caused by the lack of negative selection. We therefore used B6.K hematopoietic stem cells (HSCs) to reconstitute lethally irradiated gp250 SC mice (B6.K > gp250 SC; *Figure 2—figure supplement 1C*). In B6.K > gp250 chimeras, radiation-resistant epithelial cells presented gp250 as the only ligand to promote positive selection, while B6.K bone marrow-derived antigen presenting cells presented a full peptide repertoire to restore negative selection. After a 12-week reconstitution, in B6.K > gp250 chimeras, the frequencies of Vβ5^+, Vβ11^+ and Vβ12^+ were indistinguishable when compared with B6.K > B6.K chimeras (*Figure 2—figure supplement 1C,D*). The CD8^+ T cell population in gp250 SC mice was similar to the size in B6.K mice (*Figure 2H*), and the Vβ usages of CD8^+ T cells in B6.K > gp250 chimeras were comparable to the frequencies in B6.K > B6.K mice (*Figure 2—figure supplement 1D*). The data reflected MHC class I-mediated positive selection was relatively intact, given no alternation was made on MHC class I-restricted peptide presentation.

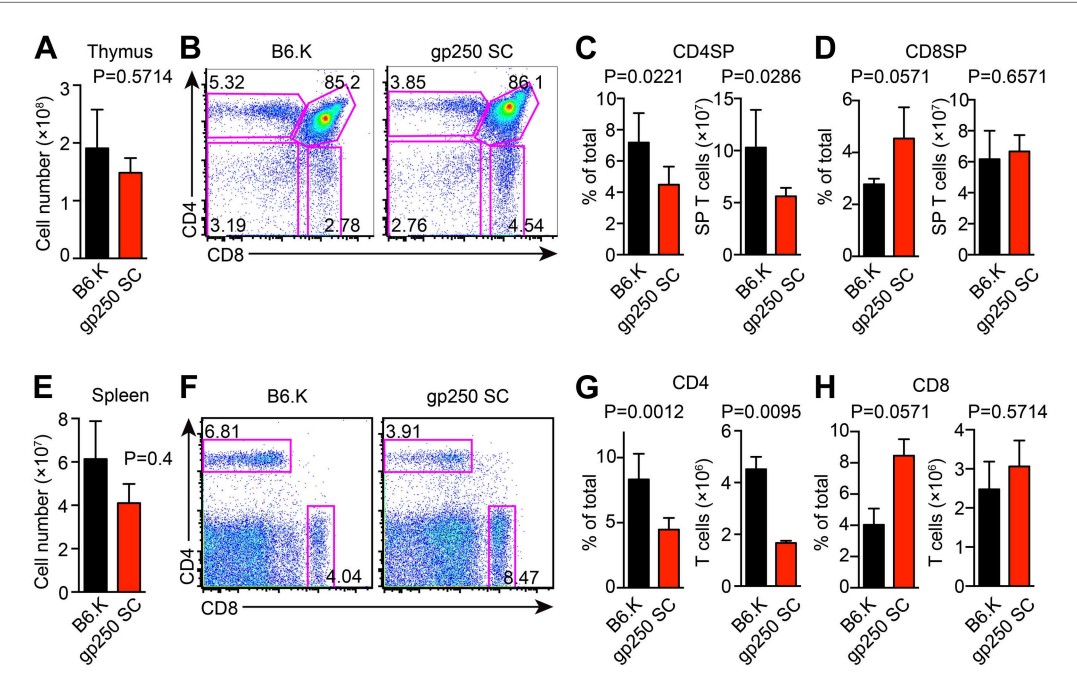

**Figure 2**. The single gp250/I-E$^k$ ligand positively selects a significant population of CD4$^+$ T cells in gp250 SC mice. (**A**) Total number of thymocytes in B6.K or gp250 SC mice. (**B**) Plot of thymocytes from B6.K or gp250 SC mice. (**C** and **D**) Quantification by percentage of CD4SP (**C**) or CD8SP (**D**) thymocytes, and of total number of CD4SP (**C**) or CD8SP (**D**) in B6.K or gp250 SC mice are shown in the bar graph. (**E**) Total number of splenocytes in B6.K or gp250 SC mice. (**F**) Plot of splenocytes from B6.K or gp250 SC mice. (**G** and **H**) Quantification by percentage of CD4$^+$ (**G**) or CD8$^+$ (**H**) T cells and of total CD4$^+$ (**G**) or CD8$^+$ (**H**) T cell numbers in the spleens of B6.K or gp250 SC mice are shown in the bar graph. (**A**–**H**) The data are representative of three experiments (B6.K, n = 7; gp250 SC, n = 6; mean ± SD; two-tailed Mann-Whitney test).

The following figure supplements are available for figure 2:

**Figure supplement 1**. TCR Vβ usages of peripheral T cells in gp250 SC mice.

Together, the data show that the single gp250/I-E$^k$ ligand was capable of positively selecting a large population of CD4$^+$ T cells, which then could be further analyzed in detail.

## The gp250-mediated positive selection greatly expands the MCC-specific CD4$^+$ T cell population

We used MCC-tetramers to identify MCC-specific population in CD4SP thymocytes and peripheral CD4$^+$ T cells. The gp250-mediated positive selection dramatically expanded the MCC-specific CD4$^+$ T cell response (*Figure 3*). In gp250 SC mice, the MCC tetramer$^+$ CD4$^+$ population showed a threefold increase in the periphery and a twofold increase in the thymus, compared with the population in B6.K mice (*Figure 3A,B*). Considering that the size of the CD4$^+$ T cell population in gp250 SC mice was around 40% of the CD4$^+$ T cell population in B6.K mice (*Figure 2*), the augmented MCC-tetramer$^+$ frequency was even more impressive. While T cell populations specific for other foreign antigens very likely became smaller, T cells specific for MCC increased in percentage, or at least stayed comparable to T cells selected by diverse self-peptide repertoire in terms of the absolute number. Indeed, the MCC tetramer$^+$ population was uniquely expanded in the gp250-selected CD4$^+$ T cell repertoire, as the frequency of the hemoglobin antigen (Hb) tetramer-bound CD4$^+$ T cell population was equivalent, if not slightly decreased (*Figure 3C,D*). To show that the increased MCC tetramer$^+$ population was the product of gp250-mediated positive selection not due to the lack of negative selection, we generated radiation bone marrow chimeras (*Figure 3E,F*). We used B6.K hematopoietic stem cells (HSCs) to reconstitute lethally irradiated gp250 SC mice (B6.K > gp250 SC). The B6.K HSC-reconstituted B6.K

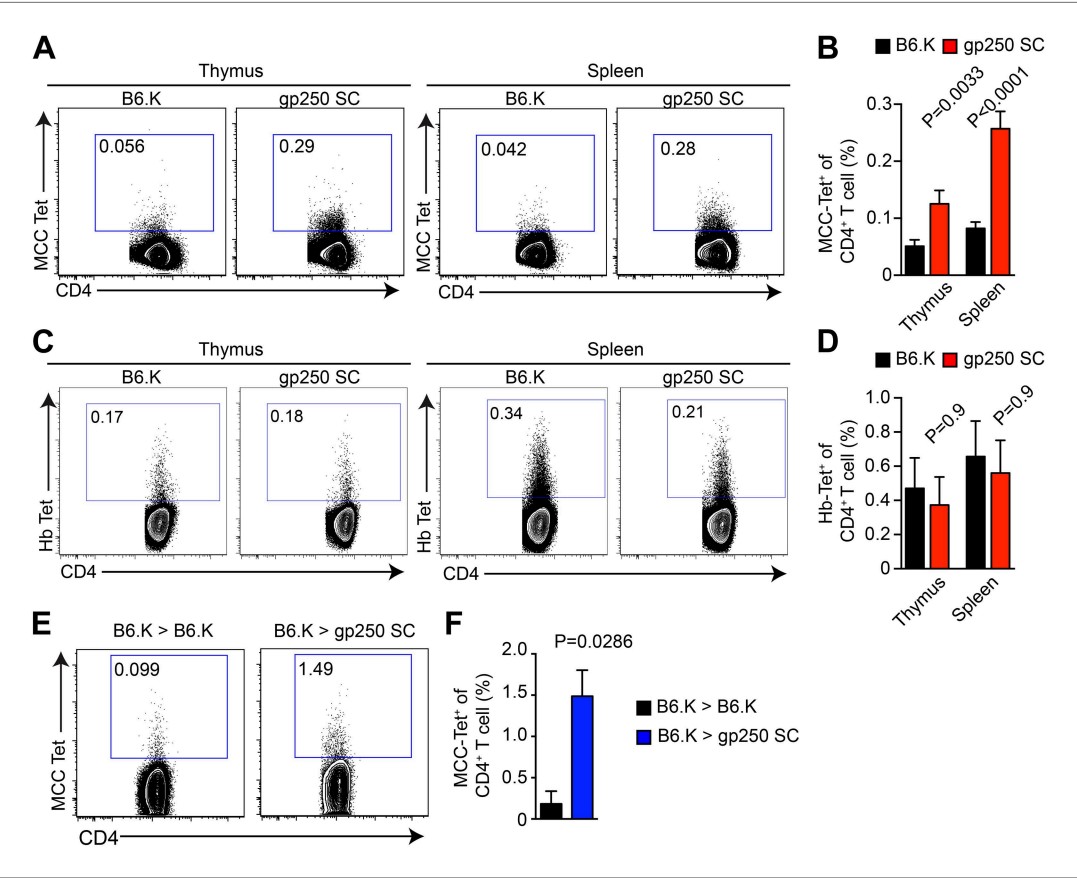

**Figure 3**. gp250-mediated positive selection greatly expands MCC-tetramer+ CD4+ T cells. (**A** and **B**) The frequency of MCC-tetramer+ CD4SP thymocytes or peripheral CD4+ T cells in B6.K or gp250 SC mice. The plot was gated on the live CD4+CD3+CD8– population. The data are representative of at least five experiments. Bar graph shows the summary (**B**; thymus, n = 12; spleen, n = 17; mean ± SD; two-tailed Mann–Whitney test). (**C** and **D**) The frequency of Hb-tetramer+ CD4SP thymocytes or peripheral CD4+ T cells in B6.K or gp250 SC mice. The plot was gated on live CD4+CD3+CD8– population. The data are representative of at least three experiments. (**D**): n = 3; mean ± SD; two-tailed Mann–Whitney test. (**E** and **F**) The frequency of MCC-tetramer+ peripheral CD4+ T cells in B6.K or gp250 SC bone marrow chimeras reconstituted with B6.K bone marrows. (**F**): n = 4; mean ± SD; two-tailed Mann–Whitney test.

bone marrow chimeras served as the controls (B6.K > B6.K). After a 12-week reconstitution, the frequency of MCC tetramer+ CD4+ T cells was 1.5% in B6.K HSC-reconstituted gp250 SC mice (*Figure 3E,F*). Compared with the frequency of MCC tetramer+ population in B6.K HSC-reconstituted B6.K mice, which was about 0.18%, gp250-mediated positive selection enriched MCC tetramer+ cells more than eightfold (*Figure 3E,F*). The B6.K HSC-reconstituted gp250 chimeras suggested the augmented MCC tetramer+ population in gp250 SC mice was not due to the lack of negative selection. Taken together, our data showed that gp250-mediated positive selection enriched for MCC-specific CD4+ T cells.

## The gp250-selected MCC tetramer+ CD4+ population is dominated by Vα11+

The polyclonal MCC-specific T cell memory responses are strongly dominated by Vα11+ CD4+ T cells; however, the basis of this is not known (*Winoto et al., 1986*; *Engel and Hedrick, 1988*; *McHeyzer-Williams and Davis, 1995*; *McHeyzer-Williams et al., 1999*; *Mikszta et al., 1999*; *Malherbe et al., 2004*; *Newell et al., 2011*). We hypothesized that recognition of a positively selecting ligand may influence a specific antigen response through optimizing the selection of dominant TCRs. We therefore included the TCR Vα11 staining in MCC-tetramer experiments to examine the Vα11+ TCR expression

in gp250-selected MCC-specific CD4+ T cells (*Figure 4A,B*). In B6.K CD4+ T cells, the MCC tetramer+ populations that used Vα11− vs Vα11+ were about equal, whereas in gp250 SC mice, the vast majority of MCC tetramer+ population expressed Vα11+ (*Figure 4A,B*). It was the increased Vα11+ MCC tetramer+ cells that specifically contributed to the enlarged MCC tetramer+ population in gp250 SC mice, since the frequency of Vα11− MCC tetramer+ population in gp250 SC mice was comparable with the frequency in B6.K mice (*Figure 4B*). The exclusively enlarged Vα11+ MCC tetramer+ population in gp250 SC mice suggested that gp250/I-E$^k$ specifically favored the positive selection of Vα11+ TCR, thus resulting in a T cell repertoire profile with the enhanced MCC tetramer-bound population. The MCC tetramer+ T cells were responsive to MCC, as the sorted MCC tetramer+ CD4+ T cells upregulated CD69 while stimulated overnight with MCC-pulsed APC in vitro (*Figure 4C,D*).

## The gp250/I-E$^k$ skews the selection of TCRs exhibiting conserved MCC-reactive features

The TCR Vβ usages in Vα11+-driven MCC-specific response exhibit a highly organized immunodominance hierarchy. On day 7 post immunization, about 50% of MCC-reactive T cells were Vα11+Vβ3+, and

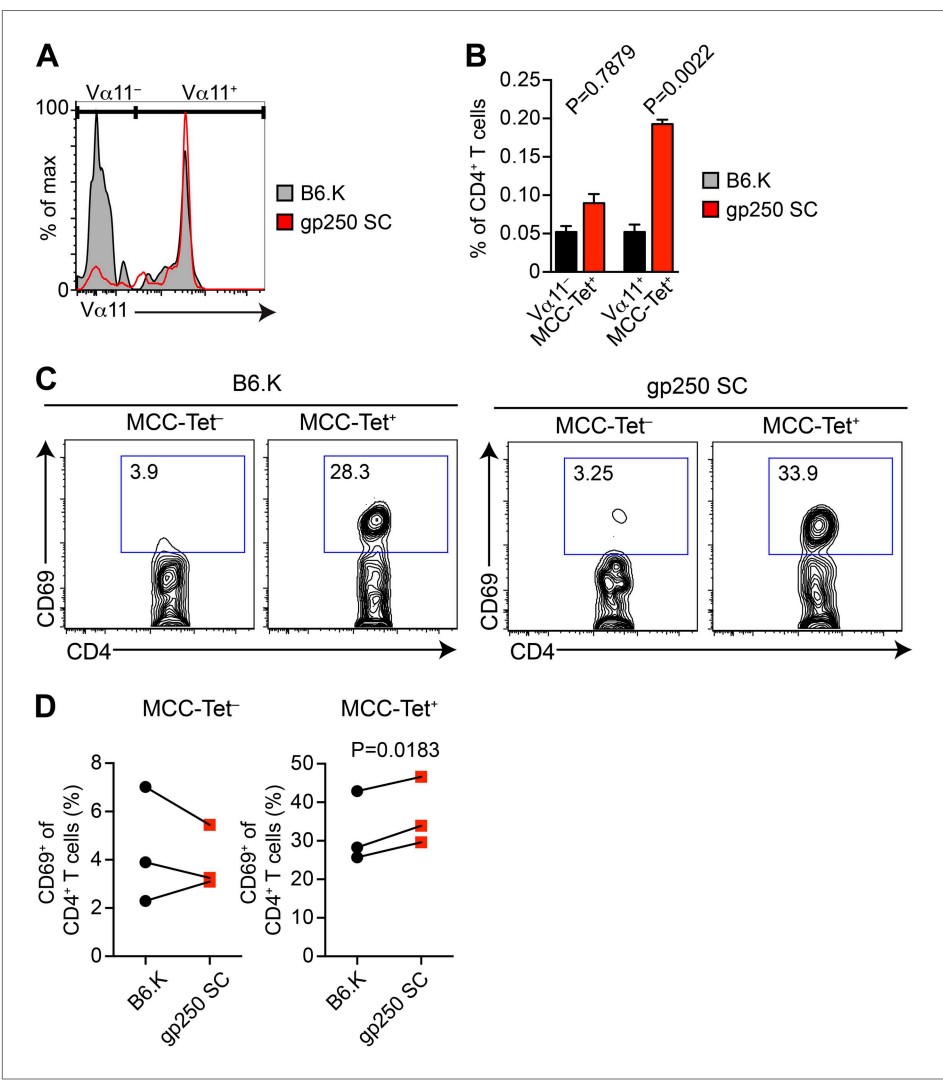

**Figure 4**. The majority of gp250-selected MCC-specific CD4+ T cells express Vα11+ TCR. (**A** and **B**) The Vα11 expression of MCC-tetramer+ peripheral CD4+ T cells in B6.K or gp250 SC mice. (**B**) n = 6; mean ± SD; two-tailed Mann–Whitney test. (**C** and **D**) The CD69 upregulation of sorted MCC-tetramer− or MCC-tetramer+ CD4+ T cells from B6.K or gp250 SC splenocytes. Data are representative of three experiments. (**D**): n = 3, paired *t* tests.

Vα11⁺Vβ6⁺, Vα11⁺Vβ8⁺, Vα11⁺Vβ14⁺ constituted the remaining 50% MCC-reactive T cells. Observing gp250-mediated selection promoted the MCC-specific CD4⁺ T cells toward the usage of Vα11⁺, we next examined the selection of MCC-preferred dominant and subdominant TCR pairs in gp250 SC mice (*Figure 5*). The Vα11⁺-expressing CD4⁺ T cells significantly increased in the thymus and periphery in gp250 SC mice (*Figure 5A,B*), suggesting gp250/I-Eᵏ actively skewed the post-selection TCR usages

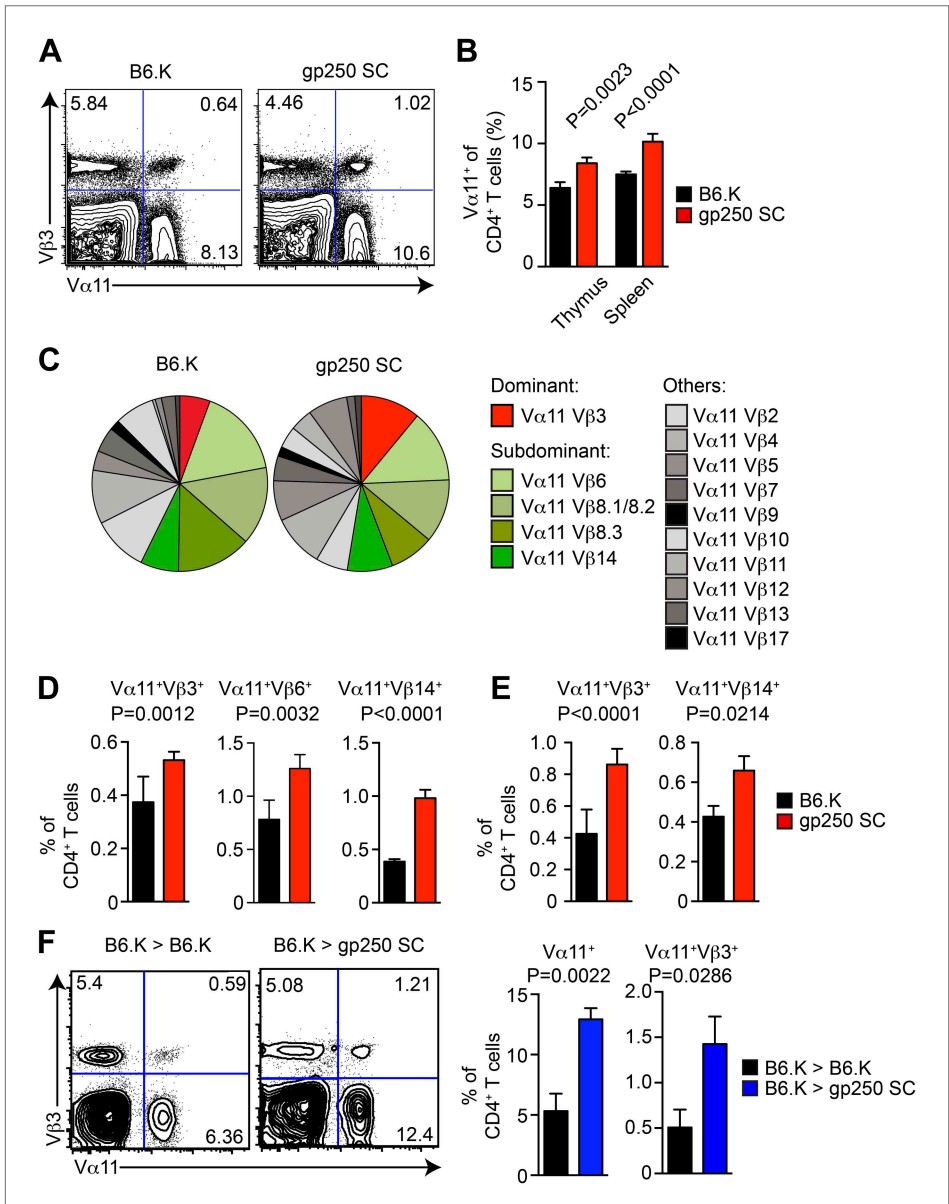

**Figure 5**. The gp250 self-peptide skews the selection of MCC-reactive preferred TCR pairs. (**A**) The Vα11 and Vβ3 expression of thymic CD4SP or peripheral CD4⁺ T cells in B6.K or gp250 SC mice. The data are representative of three experiments. (**B**) The quantification by percentage of Vα11⁺ CD4SP thymocytes or CD4⁺ peripheral T cells in B6.K or gp250 SC mice. n = 7; mean ± SD; two-tailed Mann–Whitney test. (**C**) The distribution of Vβ usages among Vα11⁺ CD4SP thymocytes or CD4⁺ peripheral T cells in B6.K or gp250 SC mice. The pie chart was plotted with the mean of three experiments (n = 7). (**D** and **E**) The quantification of increased dominant and subdominant MCC-reactive TCR pairs of CD4SP thymocytes or CD4⁺ peripheral T cells in B6.K or gp250 SC mice. n = 7; mean ± SD; two-tailed Mann–Whitney test. (**F**) The Vα11 and Vβ3 expression of thymic CD4SP or peripheral CD4⁺ T cells in B6.K or gp250 SC bone marrow chimeras reconstituted with B6.K HSC cells. The data are representative of three experiments. n = 4; mean ± SD; two-tailed Mann–Whitney test.

to influence antigen specificities. While examining the TCR Vβ expression among gp250-selected Vα11+ CD4+ T cells, MCC-preferred dominant pair Vα11+Vβ3+ increased about 1.4-fold in thymic CD4SP and at least twofold in periphery (*Figure 5C–E*). Certain subdominant MCC-preferred TCRs also increased (*Figure 5C–E*). Interestingly, for some Vβs, found commonly to pair with Vα11 in normal mice (such as Vβ2 and Vβ4), the frequencies were unchanged (*Figure 5C*). This observation argued that gp250 specifically promoted the selection of MCC-specific TCR usages, and was not merely functioning as a good selecting ligand for all Vα11+. The increased frequency of Vα11+Vβ3+ did not result from the absence of negative selection in gp250 SC mice (*Figure 5F*). While reconstituted lethally irradiated gp250 SC mice with B6.K HSC cells, the negative selection in gp250 SC mice was restored due to the presence of B6.K HSC-derived antigen presenting cells in the thymus. After a 12-week reconstitution, Vα11+ CD4+ peripheral T cells in B6.K > gp250 SC bone marrow chimeras were about 2.5-fold larger than the size in B6.K > B6.K chimeras (*Figure 5F*). The Vα11+Vβ3+ population increased to approximately threefold in B6.K > gp250 SC chimeras, compared with the population in B6.K > B6.K chimera mice as well (*Figure 5F*). The bone marrow reconstitution data clearly supported gp250-mediated positive selection was the driving force to cause the enlarged MCC-specific CD4+ population.

## gp250 self-peptide favors positive selection of MCC-reactive conserved CDR3 features

In addition to the preferred TCR Vα and Vβ usages, MCC-responsive CD4+ T cells also showed strikingly conserved CDR3 features, including the preferred four CDR3 residues: glutamic acid at CDR3 α89 (α89E), serine at α91 (α91S), asparagine at CDR3 β97 (β97N), and alanine/glycine at β99 (β99A/G) (*McHeyzer-Williams et al., 1999*; *Newell et al., 2011*). The crystal structures of TCR:pMHC complexes have provided the structural explanation for these conserved CDR3 residues for each may form salt bridge or hydrogen bond with MCC TCR contact residues at P3 (tyrosine), P5 (lysine), and P8 (threonine) (*Newell et al., 2011*). To elucidate the amino acid residues at these specific CDR3 positions, we performed single cell repertoire analysis and large-scale bulk CDR3 sequence studies (*Figure 6*).

We single cell sorted Vα11+Vβ3+ peripheral CD4+ T cells from a gp250 SC mouse for single cell TCR analysis. We successfully recovered 38 sequences of TCR Vα11 (*Table 1*), among which five also had the co-expressed Vβ3 sequences identified (*Table 2*). The Vα11+Vβ3+ populations were also sent for large-scale bulk CDR3α and CDR3β sequencing to extensively explore CDR3 features. We obtained about 2900 unique CDR3α and 1500 CDR3β sequences in gp250-selected Vα11+Vβ3+ CD4+ T cells (*Table 3*). We found AND TCR and its closely related TCR: 5cc7 and 226 TCRs (*Tables 1 and 3*). In bulk population sequence analysis, the majority of published MCC-reactive TCR CDR3α sequences were identified in our dataset, and we found six of the published CDR3β sequences (*Table 3*) (*Hedrick et al., 1988*; *McHeyzer-Williams et al., 1999*).

Analysis of CDR3α residues in the bulk populations revealed that gp250-mediated positive selection significantly enriched the usage of serine at CDR3 α91 in Vα11+Vβ3+ CD4+ T cells (*Figure 6A,B*), which hydrogen bonds to the MCC peptide tyrosine residue at P3[28]. In B6.K mice, about 20% of Vα11+Vβ3+ CD4+ T cells expressed serine at CDR3 α91, whereas in gp250 SC mice, the frequency was increased 1.5-fold to become about 30% (*Figure 6A,B*). We observed a similar 1.5-fold increase of serine at α91 in single cell repertoire analysis compared to the published data (*McHeyzer-Williams et al., 1999*) (*Figure 6B*). However, gp250-mediated selection had little impact on skewing the amino acid residue at CDR3 α89, as α89E usages of Vα11+Vβ3+ CD4+ T cells were comparable with the frequency seen in B6.K (*Figure 6A,B*). The large-scale bulk CDR3β sequencing showed gp250-mediated positive selection dramatically escalated the usages of β97N, slightly increased the frequency of β99A, and had no effect on the frequency of β99G (*Figure 6C,D*). Nevertheless, B6.K CDR3β features were highly similar to the published CDR3's (*McHeyzer-Williams et al., 1999*; *Figure 6C,D*). The CDR3 β97N residue was hypothesized to hydrogen bond with MCC P8 backbone carbonyl and MCC P8 threonine side-chain hydroxyl groups (*Newell et al., 2011*). The significantly increased usages of CDR3 α91S and β97N in gp250-selected Vα11+Vβ3+ T cells may also help explain the augmented MCC-tetramer bound population in gp250 SC mice.

Altogether, our data changed our current understanding of the relationship between positively selecting self-peptides and post-selection T cell repertoire (*Figure 7*). With the well-defined MCC's highly organized immunodominance hierarchy, we discovered gp250's active role in promoting the positive selection of good MCC responders. The self-peptide gp250 enhanced the selection of

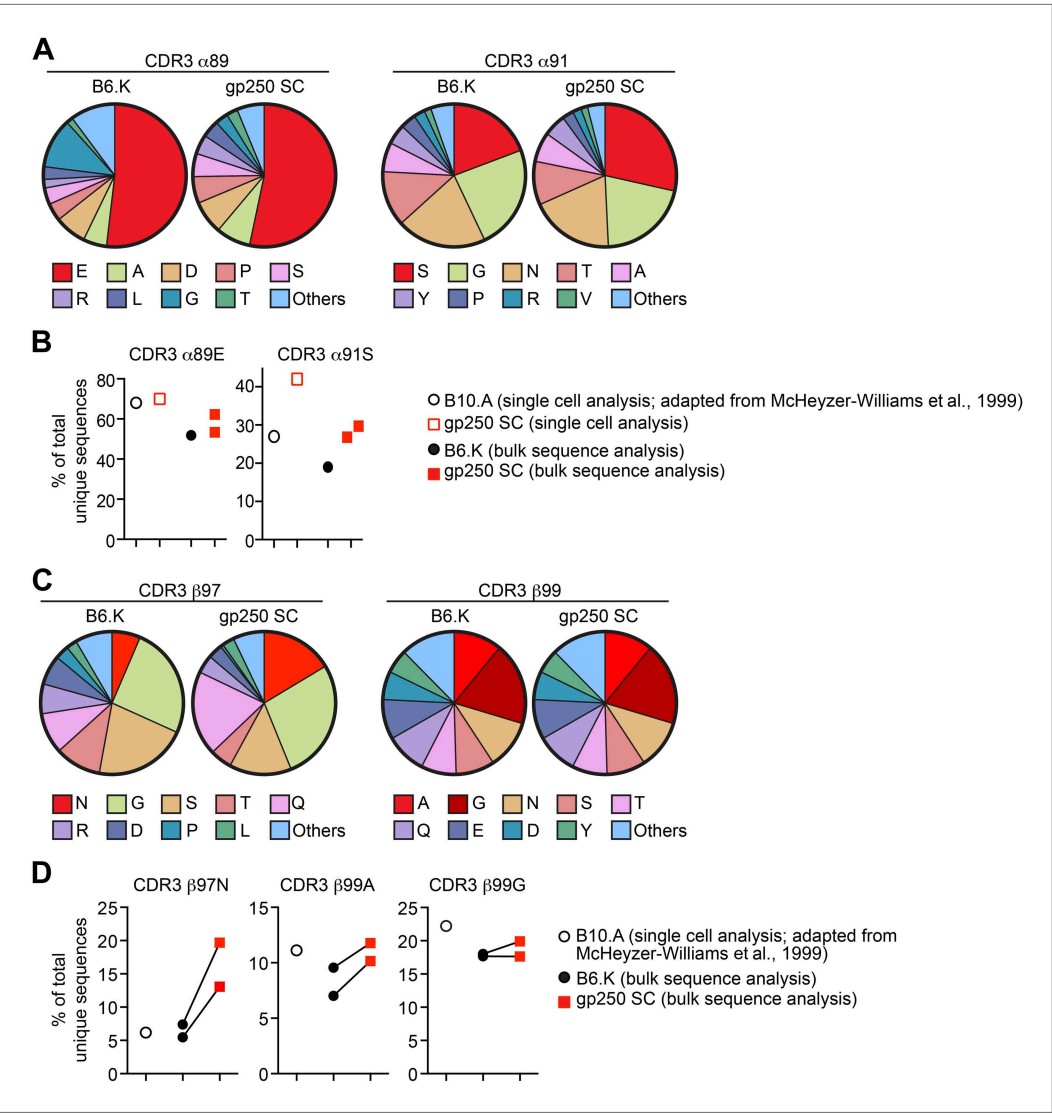

**Figure 6**. The gp250 self-peptide skews the selection of TCRs exhibiting conserved MCC-reactive features. (**A**) The pie chart of amino acid usages at CDR3 α89 and α91 of bulk-sorted peripheral Vα11⁺Vβ3⁺ CD4⁺ T cells in B6.K or gp250 SC mice. The gp250 SC data were obtained by single cell repertoire analysis. (**B**) The frequency of MCC-reactive preferred CDR3α amino acids, α89N and α91E from gp250-selected peripheral Vα11⁺Vβ3⁺ CD4⁺ T cells in single cell analysis (left) or bulk population analysis (right). (**C**) The pie chart of amino acid usages at CDR3 β97 and β99 from bulk-sorted peripheral Vα11⁺Vβ3⁺ CD4⁺ T cells in B6.K or gp250 SC mice. The data were representative of two independent experiments. (**D**) The frequencies of CDR3 β97N, β99A, and β99E of Vα11⁺Vβ3⁺ CD4⁺ T cells in B6.K or gp250 SC mice. (**B** and **D**) The single cell B10.A data were plotted based on *McHeyzer-Williams et al. (1999)* published data.

Vα11⁺Vβ3⁺ T cells, which were the dominant TCR pair against MCC. These gp250-selected Vα11⁺Vβ3⁺ T cells also exhibited conserved MCC-reactive features, including the specific amino acid usages at CDR3 α91 and β97. Our studies showed an ideal positively selecting self-peptide may be a key determinant to optimize immune response against a specific antigen, by enhancing the positive selection of good responders.

## Discussion

Immunodominance describes a common phenomenon of T cell immune responses. Yet the role of positive selection in immunodominance has not been established. In this study, we reported that CD4⁺

**Table 1.** Examples of CDR3α sequences of Vα11⁺Vβ3⁺ CD4⁺ T cells from gp250 SC mice in the single cell repertoire analysis

| CDR3α sequences | Jα usages | Frequency | Notes |
|---|---|---|---|
| CAAEASSGSWQLIF | TCRA17 | 21% (8/38) | Also published in *McHeyzer-Williams et al. (1999)* |
| CAAEANSGTYQRF | TCRA11 | 7.8% (3/38) | |
| CAAEAGGGSGGKLTL | TCRA36 | 5.2% (2/38) | |
| CAAEASSGQKLVF | TCRA13 | 5.2% (2/38) | AND TCR CDR3α; also published in *McHeyzer-Williams et al. (1999)* |
| CAAEPPHANTGANTGKLTF | TCRA44 | 5.2% (2/38) | |
| CAANTGNYKYVF | TCRA33 | 5.2% (2/38) | |
| CAADRSNNRIFF | TCRA24 | 5.2% (2/38) | |
| CAAEPSSFSKLVF | TCRA24 | 2.6% (1/38) | |
| CAAESSNMGYKLTF | TCRA8 | 2.6% (1/38) | |
| CAARSSNTNKVVF | TCRA8 | 2.6% (1/38) | |

**Table 2.** The single cell repertoire analysis of Vα11⁺Vβ3⁺ CD4⁺ T cells from gp250 SC mice

| CDR3α sequences | Jα usages | CDR3β sequences | Jβ usages |
|---|---|---|---|
| CAADRSNNRIFF | TCRA17 | CASSKQANSYNSPLYF | Jβ2.4 |
| CAAESSNMGYKLTF | TCRA24 | CASSGDGRGNTLYF | Jβ1.8 |
| CAARSSNTNKVVF | TCRA27 | CASSLWANTGQLYF | Jβ2.2 |
| CAAEANSGTYQRF | TCRA11 | CASSLLHKQYF | Jβ2.1 |
| CAAEASSGSWQLIF | TCRA17 | CASSPGTQNTLYF | Jβ2.4 |

T cells specific for a defined antigen are positively selected by a common self-peptide. The self-peptide gp250 positively selected AND TCR, supporting our previous in vitro study (*Lo and Allen, 2013*), and also selected many other TCRs that have been reported as good MCC responders (*Hedrick et al., 1988*; *McHeyzer-Williams et al., 1999*). Our data reveal for the first time that positive selection may influence the TCR V(D)J preference and specific CDR3 junction sequences, and thus shape T cell response to a specific antigen. Our studies therefore offer direct evidence and a new perspective on how to generate a post-selection CD4⁺ T cell repertoire with desired antigen specificities, through engineering the positively selecting self-peptide.

The gp250-selected CD4⁺ T cell repertoire analysis uncovered several very interesting results regarding the specificity of recognizing a positively selecting self-peptide, as well as the relationship between a positively selecting self-peptide and post-selection T cells. In the bulk population TCR repertoire analysis, the CDR3α was found for the well-characterized AND, 5c.c7 and 226 TCRs and also for the majority of published MCC-preferred clones. These findings suggested that one selecting ligand can select as many CDR3s with conserved features for specific antigen recognition as a full self-peptide repertoire could select. Previously, the Davis laboratory and we have shown in vitro that the self-peptide for AND and 5c.c7 TCRs were mutually exclusive with each other (*Ebert et al., 2009*; *Lo and Allen, 2013*). The basis for why 5c.c7 was selected in vivo but not in vitro by gp250 probably reflects the relative inefficiency of the in vitro positively selecting assays. Our data suggest these strong MCC-reactive TCR clones may share a common positively selecting ligand, and support our hypothesis that positive selection may be the foundation of T cell immunodominance.

Our finding that gp250 was sufficient to generate a reduced but still a large number of mature CD4⁺ T cells, with a full but skewed range of TCR usages, was similar to the observations in other single peptide mice, such as Eα-, CD22-, or Rab-mediated positive selection (*Ignatowicz et al., 1996*, *1997*;

**Table 3.** The CDR3 sequences of bulk-sorted Vα11+Vβ3+ CD4+ T cells from gp250 SC mice that overlap with the published data from normal mice (*Hedrick et al., 1988*; *McHeyzer-Williams et al., 1999*)

| Published CDR3α (*Hedrick et al., 1988; McHeyzer-Williams et al., 1999*) | Current study | | |
|---|---|---|---|
| | **B6.K** | **SC** | **Notes** |
| | **3623** | **2906** | **Unique sequences** |
| CAAEASSGQKLVFG | + | + | AND TCR |
| CAAEPSSGQKLVFG | + | + | 226 TCR |
| CAAEASGSWQLIFG | + | + | |
| CAAEASNTNKVVFG | + | + | 5c.c7 TCR |
| CAAEGSNTNKVVFG | + | | |
| CAAEASAGNKLTFG | + | + | |
| CAAASSGSWQLIFG | + | + | |
| CAAEAGSNAKLTFG | + | + | |
| CAAEASNNNAPRFG | + | | |
| CAAEAASLGKLQFG | + | + | |
| CAAEASSGSWQLIFG | + | + | |
| CAAEASNYNVLYFG | + | + | |
| CAAEASSSFSKLVFG | + | + | |
| CAAEASNMGYKLTFG | + | + | |
| CAAETGGYKVVFG | + | + | |
| CAAEANYNQGKLIFG | + | + | |
| CAAEAGSGTYQRFG | + | + | |
| CAGLSGSFNKLTFG | + | + | |
| CAAEGNTGNYKYVFG | | + | |
| CAAEEGNMGYKLTFG | + | + | |
| CAATSSGQKLVFG | + | | |

| Published CDR3β (*Hedrick et al., 1988; McHeyzer-Williams et al., 1999*) | Current study | | |
|---|---|---|---|
| | **B6.K** | **SC** | **Notes** |
| | **2821** | **1583** | **Unique sequences (mean)** |
| CASSLNSANSDY | + | | AND, 5c.c7, 226 TCRs |
| CASSLNNANSDY | + | + | |
| CASSLSTSQNTLYF | + | | |
| CASSLQGTNTEVFF | + | + | |
| CASRLGQNTLYF | | + | |
| CASSLGASAETLYF | | + | |

*Barton and Rudensky, 1999*; *Barton et al., 2002*) (*Figure 7*). However, in those previously published studies, the specificity of post-selection repertoire was only examined at a cursorily level by stimulating the CD4+ T cells with random known antigens. The gp250 SC mice, however, successfully provided some key insights toward the relationship between positively selecting self-peptide and post-selection CD4+ T cells. Thanks to the well-studied MCC polyclonal T cell responses, we were thus able to discover some unique features of positive selection (*McHeyzer-Williams and Davis, 1995*; *McHeyzer-Williams et al., 1999*; *Mikszta et al., 1999*; *Malherbe et al., 2004*). We found positively selecting self-peptide may favor some unique CDR3 features, which may serve as a good primary and memory responder toward a specific antigen in the periphery.

Consistent with previous studies, the gp250 peptide was able to select a large number of CD4+ T cells. This observation makes sense when one considers that the number of selecting self-peptides presented in

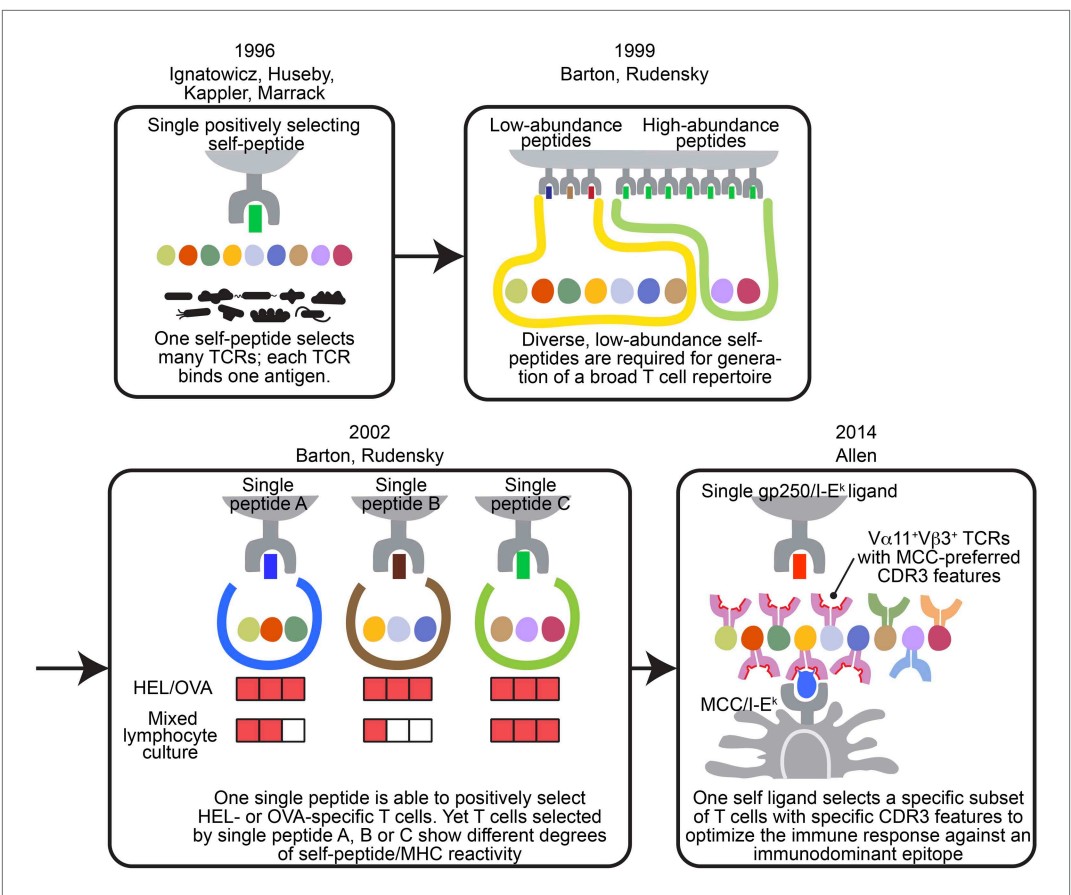

**Figure 7**. The evolving concepts of the relationship between positively selecting self-peptide and post-selection T cell repertoire. The figures show the progress of our understanding about the relationship between thymic self-peptides and post-selection CD4+ T cell repertoire, and are arranged in chorological order. The first finding from late 90' revealed that one positively selecting self-peptide was capable of selecting many different TCRs (*Ignatowicz et al., 1996*; *Ignatowicz et al., 1997*; *Liu et al., 1997*; *Huseby et al., 2005*). A few years later, Barton and Rudensky proposed that even though one self-peptide may select many TCRs, a diverse low abundant self-peptide repertoire is required to generate a full TCR repertoire (*Barton and Rudensky, 1999*). Subsequently, with the publication of a series of single chain mice, the model was further refined because even though the TCR repertoire selected by one single ligand was not reduced significantly enough to allow an visualization of a repertoire 'hole', the postselection T cells in each single chain mouse showed different degrees of self-peptide reactivity in mixed lymphocyte culture (*Barton et al., 2002*). These studies showed that a diverse collection of positively selecting self-peptides were necessary to generate a full T cell repertoire (*Barton and Rudensky, 1999*; *Barton et al., 2002*). This present study involving the gp250 self-peptide showed increased frequency of Vα11+Vβ3+ CD4+ T cells with MCC-preferred CDR3 features in gp250 SC mice. The Vα11+Vβ3+ CD4+ T cells with specific CDR3 features were the dominant responders to promote MCC primary and memory responses. Therefore, gp250's selecting capability provided a possible explanation to elucidate Vα11+ TCR-driven MCC immunodominance: a positively selecting self-peptide may favor the selection of TCR pairs and CDR3 features that were specific for MCC responses.

cortical thymic epithelial cells is smaller than the size of mature CD4+ T cell repertoire (*Lo and Allen, 2013*). Therefore, one positively selecting self-peptide has to select a large number of CD4+ T cell with diverse, unrelated antigen specificities, to be able to generate a full CD4+ T cell repertoire. The thymic development of regulatory T cells is also dependent upon self-peptides, and requires the strong interaction with these self-peptide MHC ligands (*Gottschalk et al., 2010*; *Ohkura and Sakaguchi, 2010*; *Corse et al., 2011*; *Ohkura et al., 2013*). However, our preliminary analysis of CD25 expression of CD4SP in the gp250 SC mice did not reveal any significant differences from B6.K mice. The comparable regulatory T cell frequencies in gp250 SC mice and B6.K mice suggested that merely altering the self-peptide diversity was

not sufficient to manipulate the affinity threshold for regulatory T cell selection. The finding was perhaps not surprising, given the potential TCR diversity through CDR3 recombination mechanism is nearly infinite, and both the TCR and the self-peptide determine the strength of interaction.

Our data showed gp250 could select many different Vα11 TCRs. How could a peptide influence the selection of many different Vα chains, with a range of CDR3α sequences? The general binding mode of TCR to pMHC has the CDR1 and CDR2 contacting the MHC, and the CDR3 contacting the peptide. One explanation comes from the crystal structure of the 226 TCR bound to MCC/I-E[k28]. The side chain of the Arg29 residue of the Vα11 CDR1 makes a hydrogen bond with the backbone carbonyl oxygen of the P3 residue (*Newell et al., 2011*). Thus gp250 could position the P3 carbonyl oxygen ideally to form this hydrogen bond, and thus favoring Vα11 TCRs. Further structural studies will be needed to reveal the molecular basis of how a single peptide can select defined TCRs.

## Materials and methods

### Mice

B6.K (H-2$^k$) (JAX 001148) and B6 MHC class II$^{-/-}$ (JAX 003584) mice were obtained from the Jackson Laboratory. The B6 CD75$^{-/-}$ mouse line was a gift from the Kappler/Marrack laboratory (*Bikoff et al., 1995*). All mice were bred and housed in specific pathogen-free conditions of the animal facility at the Washington University Medical Center. All the use of laboratory animals was approved and done in accordance with the Washington University Division of Comparative Medicine guidelines.

### Flow cytometry

Fluorescence-conjugated antibodies from commercial sources were as follows: fluorescein isothiocyanate-, allophycocyanin- or allophycocyanin–indodicarbocyanine-conjugated anti-CD4 (GK1.5; BioLegend, San Diego, CA); phycoerythrin–indodicarbocyanine-conjugated anti-CD8 (53-6.7; BioLegend); fluorescein isothiocyanate-, or phycoerythrin-conjugated I-E$^k$ (17-3-3; BioLegend); allophycocyanin- or phycoerythrin–cy7-conjugated anti-CD69 (1-11.2F3; BioLegend); LIVE/DEAD Fixable Blue Dead Cell Stain (Invitrogen, Carlsbad, CA); or fluorescein isothiocyanate-conjugated or biotinylated anti-Vα11 (RR8-1; BD Pharmingen, Franklin Lakes, NJ). Antibodies against the panel of TCR Vβ chains were purchased from BD Pharmingen (557004). All samples were analyzed on a FACSCalibur, FACSLSRII, or FACSAria (BD) and data were analyzed with FlowJo software (TreeStar, Ashland, OR).

### Generation of gp250 SC mice

We used a construct based on the hemoglobin peptide (Hb$_{64-76}$) covalently linked to the I-E$^k$ β chain with a flexible linker (a generous gift of J Kappler). We replaced the hemoglobin DNA sequences with the gp250 peptide by PCR to create the gp250/I-E$^k$β construct. The gp250/I-E$^k$ construct was subsequently subcloned into the EcoRI sites of pDOI-5 vector. A BglI linearized fragment was isolated and used to inject the pronuclei of fertilized B6 oocytes to obtain the gp250/I-E$^k$ mouse. We screened 374 mice and obtained one founder. The founder was crossed to I-E$^k$ α chain transgenic mice (*Griffiths et al., 1994*), and subsequently bred onto the MHCII$^{-/-}$ and CD74$^{-/-}$ background. All of the experiments reported here used the gp250 single chain mice on MHCII$^{-/-}$ and CD74$^{-/-}$ background.

### Flow cytometric analysis of mouse TCR Vβ repertoire

Thymocytes or splenocytes from gp250 SC mice or B6.K mice were stained with a panel of FITC-conjugated antibodies against a panel of antibodies against mouse TCR Vβ chains (557004; BD Pharmingen), along with APC-CD4, PE-Vα11, and APC-Cy7-CD8.

### MCC tetramer staining

APC-labeled I-E$^k$ tetramers in complex with MCC, or CLIP were obtained from the NIH facility. All washes and incubations were done in modified RPMI-1640 containing 2% FCS and 0.075% sodium bicarbonate. For tetramer staining, peripheral CD4$^+$ T cells were enriched by CD4$^+$ MACS beads (Miltenyi Biotech, Germany), and CD4SP thymocytes were enriched by negative selection of CD8$^+$ MACS beads (Miltenyi Biotech). Enriched CD4$^+$ peripheral T cells or CD4SP thymocytes were stained with APC-conjugated MCC tetramer or PE-conjugated Hb tetramer on ice for about 1.5–2 hr. The APC-conjugated or PE-conjugated CLIP tetramers were used as a negative control. In the case of MCC tetramer staining, the cells were also stained for APC/Cy7-conjugated CD4, PE/Cy7-conjugated CD8, and Pacific blue-conjugated CD3 antibodies. Cells positive for LIVE/DEAD Fixable Blue Dead Cell Stain were gated out of the analysis.

## Bone marrow chimeras

B6.K or gp250 SC hematopoietic stem cells were enriched by c-kit MicroBeads (#130-091-224; Miltenyi Biotech), stained for c-kit and Sca-1, and sorted out c-kit$^+$Sca-1$^+$ population on an AriaII. Hematopoietic stem cells were resuspended at $10^4$ cells per 50 µl PBS and were adoptively transferred into lethally irradiated B6.K or gp250 SC mice by retro-orbital injection on day 0 (at least $10^4$ hematopoietic stem cells per mouse). After a 12-week of reconstitution, cells were recovered from thymus and spleen of each recipient mouse, and analyzed by flow cytometry.

## Single cell repertoire analysis

The single cell repertoire analysis was conducted by following the protocol developed by McHezyer-William's lab (*McHeyzer-Williams et al., 1999*). Single peripheral Vα11$^+$Vβ3$^+$CD3$^+$CD4$^+$CD8$^-$ cells from gp250 SC mice were sorted into a 96-well plate on Aria II. Each single cell was sorted into the center 60 wells with the first and last well of each row serving as a negative control. Each single cell was directly sorted into a well containing 5 µl of reaction mixture: 1× Cell Direct Reaction Buffer (11753-100; Invitrogen), 0.1 µl SuperScript-III RT/Platinum Taq Mix (11753-100; Invitrogen), and 2.5 nM external primers (Va11.L2: AATCTGCAGTGGGTGCAGATTTGCTGG; Ca.2: AATCTGCAGCGGCACATTGATTTGGGA; Vβ3.L2: ATGGCTACAAGGCTCCTCTGGTA; Cb.2: CACGTGGTCAGGGAAGAA). After the single cell sort, the plate was immediately held at 50°C for 15 min, followed by 95°C for 2 min, and then 22 cycles of 95°C for 15 s and 60°C for 4 min for cDNA reaction. The cDNA products were diluted to 20 µl with molecular-grade water, and 2 µl of the first round product was used for further 10 µl amplification for TCR Vα11 and Vβ3 as previously described with the Internal Primers. For Vα11, the Internal Primers are Vα11.L3: AGATTTGCTGGGTGAGAGGAG and Cα.ext: GAGTCAAAGTCGGTGAACAGG. For Vβ3, the Internal Primers are Vβ3.1: AATCTGCAGAATTCAAAAGTCATTCAG and Cβ.3: AATCTGCAGCACGAGGGTAGCCTTTTG. The first round and second round PCR reaction were prepared and conducted at a separate bench in the lab where no AND transgenic TCR-related experiments had been previously done. After the PCR amplication, 4 µl PCR products were run on 2% and screened for the single positive bands of the right sizes. PCR products were then TOPO cloned to pCR2.1 vectors, and prepared for sequencing. The sequencing primers were as previously described (*McHeyzer-Williams and Davis, 1995*; *McHeyzer-Williams et al., 1999*).

## Large-scale bulk CDR3 analysis

The peripheral Vα11$^+$Vβ3$^+$, Vα11$^+$ Vβ6$^+$, Vα11$^+$ Vβ14$^+$ CD4$^+$ cells from B6.K or gp250 SC mice were stained and sorted on an Aria II. For TCR CDR3α analysis, RNA was isolated and used to generate cDNA as described (*Lathrop et al., 2011*). A bulk Vα11 cDNA library was generated by PCR using primers derived from published studies (*Dash et al., 2011*) and then sequenced using Illumina MiSeq technology at the Washington University Genome Sequencing Center. For TCR CDR3β repertoire, the bulk-sorted populations were pelleted at 400×*g* for 10 min and snap-frozen on dry ice. The prepared cell pellets were sent to Adaptive Biotechnologies for conducting bulk CDR3β analysis. The CDR3 amino acid usages were analyzed in Excel.

## Statistics

All data were analyzed nonparametrically by the Mann-Whitney U-test or paired *t* tests with Prism 6 software (Graph Pad). p values of less than 0.05 were considered significant.

## Acknowledgements

We thank J Kappler (National Jewish Health) for providing the pDOI-5 vector; P Marrack (National Jewish Health) for providing B6 CD74$^{-/-}$ mice; D Kreamalmeyer for maintaining the mouse colony; M White for microinjection of generating gp250 SC mice; S Horvath for peptide synthesis and purification; NIH tetramer core facility for providing MCC-, Hb- and Clip-tetramers (National Institute of Health); M McHeyzer-Williams and L McHeyzer-Williams (The Scripps Research Institute) for their help with the single cell repertoire analysis; L Turka (Massachusetts General Hospital) for his advice on large-scale bulk TCR Vβ sequencing; G Amarasinghe, C Garcia (Stanford University) and M Birnbaum (Stanford University) for the helpful discussion on structural features of recognition of MCC/I-E$^k$; C Moon, Y Wang for cryosection preparation and imaging. P Ni, D Donermeyer, O Kanagawa (Akashi City Hospital), P Ebert (Genentech) for critical reading of the manuscript and comments. Supported by the National Institutes of Health grant: AI-24157 (PMA).

# Additional information

## Funding

| Funder | Grant reference number | Author |
|---|---|---|
| National Institutes of Health | AI-24157 | Paul M Allen |

The funder had no role in study design, data collection and interpretation, or the decision to submit the work for publication.

## Author contributions

W-LL, Conception and design, Acquisition of data, Analysis and interpretation of data, Drafting or revising the article; BDS, C-SH, Acquisition of data, Analysis and interpretation of data; DLD, Conception and design, Acquisition of data; PMA, Conception and design, Analysis and interpretation of data, Drafting or revising the article

## Ethics

Animal experimentation: This study was performed in strict accordance with the recommendations in the Guide for the Care and Use of Laboratory Animals of the Washington University School of Medicine. All of the animals were handled according to approved institutional animal care and use committee (IACUC) protocols (#20110102) of the Washington University School of Medicine. The protocol was approved by the Committee on the Ethics of Animal Experiments of the Washington University School of Medicine (Permit Number: A-3381-01).

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
