## [Decision Letter]

Thank you for sending your work entitled “T cell immunodominance is dictated by the positively selecting self-peptide” for consideration at *eLife*. Your article has been favorably evaluated by a Senior editor and 2 reviewers, one of whom, Michel Nussenzweig, is a member of our Board of Reviewing Editors, and the other, Steve Hedrick, has also agreed to reveal his identity.

The Reviewing editor and the other reviewers discussed their comments before we reached this decision, and the Reviewing editor has assembled the following comments to help you prepare a revised submission.

The present paper examines the role of thymic positive selection to a single self-peptide in determining the frequency of T cells specific for a foreign peptide/MHC II that end up in the naïve CD4 repertoire; based on the work of others, they equate this with the phenomenon of immunodominance in the T cell immune response. This is a valuable piece of basic research and contributes new quantitative data to our understanding of what positive selection of the repertoire actually means. They have chosen a good model system to use in this study as a lot is known about the molecular specificity of CD4 T cell recognition of moth/pigeon cytochrome c peptides bound to I-E^k^. This allows them to see subtle but significant increases in the frequency of certain contact amino acid residues (α91S and β97N) in the naïve repertoire when only gp250/I-E^k^ self-ligand is the selecting element. Their chimera experiments show that these observations are not due to negative selection. This work we think adds significantly to the literature and should be published.

1) Our only concern about the paper is the discussion of the specificity controls involving the 5c.c7 TCR. Intriguingly, previous in vitro experiments from Mark Davis' lab showed that the major self-peptide (GP peptide) selecting that particular moth cytochrome c-specific TCR is different from the one used for the AND TCR. Thus, one would suspect that the repertoire changes seen with gp250 might not include those seen with the GP peptide and vice versa. While doing the complete crisscross set of experiments with a GP-peptide, antigen transgenic would be the ideal science to test this hypothesis, we think that would require several more years of research and would be asking far too much to be done for any journal. Instead, the authors try to argue that the TCR CDR3α signature sequence for 5c.c7 (as well as that for the 226 T cell) was not found in their single cell TCR Vα11 sampling and thus was not enriched by gp250/I-E^k^ mono-selection. However, they only sequence a total of 38 Vα11 clones, among which they found only 2 had the AND CDR3α sequence. This is not a large enough sample size to reach a significant conclusion. We therefore think they should repeat this CDR3α single cell sequencing experiment to increase their sample size to a number from which they can show statistical significance for the enrichment of AND CDR3α and still no detectable 5c.c7 and 226 CDR3α sequences.

2) In the first figure it seems that gp250 preferentially selects for CD8 T cells. Have the authors examined these cells for TCR receptor usage, and do they differ from the CD4s?

3) Please describe in more detail the B6K>gp250 chimeras and why this relates to negative selection (i.e., increases the selecting self peptides...).

4) Table 3: the authors might want to reference the initial sequence description of many of the TCR sequences presented, Science, March, 1988. The description and points made in Figure 4 are unclear. Please re-write this section. Again, in Figure 4, please make clear why this experiment restores negative selection.

5) It is not entirely clear how the present data suggest a model different from that of Barton and Rudensky. Did B and R really specify that all the T cell clones specific for a given antigen-peptide originate from a single positively selecting peptide? Or is the proposed model more of a modification of the B and R proposal?

6) The authors sequenced 38 TCR chains and did not detect the 5c.c7 or 226 TCR CDR3α chains. They conclude that the gp250 peptide does not select this TCRα sequence. To bolster this claim, they could analyze the frequency of repeat sequences and from this calculate the total number of sequences selected by gp250. Without sequencing more chains, they might be able to predict the likelihood that another unique sequence will be found. Either way they can use this analysis to comment on the possibility that they have just missed the 5c.c7 sequence.

7) Overall there are some very rough stretches of writing. This is particularly true of the Discussion. “What our data reveal for the first time, to our knowledge, is alternation of positively selecting self-peptide repertoire can lead to gain-of- function specificity of post-selection T cell repertoire.” We don't know what that means, and grammatically it is a little awkward. The following sentence including, “imprint the post-selection T cells toward the recognition of a specific antigen...” We don't think that is what the authors intend. Our interpretation of the data is that positive selection can influence the TCR V(D)J preference and specific junctional sequence usage in the T cell response to a specific antigen. Positive selection can imprint preferential VDJ sequences on an antigen response. It doesn't imprint the post-selection T cells toward the recognition of a specific antigen-that issue is not addressed in this study. We recommend a close examination of the Discussion.

In addition, this manuscript is written for a specialist in the field, and it could be made to be more suited to a general interest journal.

Overall this is a lovely study using a newly created mouse strain to probe the nuances of positive selection in the thymus, and its influence on an antigen response. With a little more analysis and editing, it would make a strong contribution to the literature on T cell development.

---

## [Author Response]

*1) and 6) Our only concern about the paper is the discussion of the specificity controls involving the 5c.c7 TCR […] Without sequencing more chains, they might be able to predict the likelihood that another unique sequence will be found. Either way they can use this analysis to comment on the possibility that they have just missed the 5c.c7 sequence*.

In our previous submission, we performed single cell repertoire analysis and sequenced 38 sequences. Among the 38 sequences, we did not find 5c.c7 TCRs. The reviewers raised an excellent point that we probably simply missed the 5c.c7, given the limited size of our single cell repertoire database. We were very intrigued by the question that whether or not gp250 can select 5c.c7 TCR, and believed the manuscript would be strengthened by having a definitive answer. We therefore collaborated with Dr. Chyi Hsieh to perform bulk population repertoire analysis of CDR3α sequences. We sorted Vα11^+^Vβ3^+^ CD4^+^ T cells from gp250 SC mice or B6.K mice, generated Vα11^+^ TCR library, and sequenced using Illumina MiSeq technology at the Washington University Genome Sequencing Center. In gp250-selected CD4^+^ T cells, we identified about 2900 unique CDR3α sequences, and we did find 5c.c7 TCR as well as AND (Table 3). More importantly, now with such a large dataset of CDR3α sequences, we were able to re-examine the MCC-preferred CDR3α features in gp250 SC and B6.K mice. Consistent with the previous observation, gp250-mediated positive selection enriched the usage of serine at CDR3 α91 residue. In B6.K mice, about 20% of Vα11^+^Vβ3^+^ CD4^+^ T cells expressed serine at CDR3 α91, whereas in gp250 SC mice, we observed about 1.5 fold increase (Figure 6). With these thousands of sequences, we believe our conclusion on the gp250-enriched MCC-preferred CDR3 features were further strengthened. We included the new data in the newly revised Figure 6 and Table 3. We also revised the Results and Discussion accordingly.

*2) In the first figure it seems that gp250 preferentially selects for CD8 T cells. Have the authors examined these cells for TCR receptor usage, and do they differ from the CD4s*?

Given that the gp250 SC mice expressed normal class I molecules, we did not anticipate any direct effect on CD8 T cell development, but there could have been some indirect effect. We now included quantification of CD8^+^ T cells displayed in absolute numbers in the newly revised Figure 2. While examining absolute cell numbers of CD8^+^ T cells, we did not find any significant difference between gp250 SC mice and B6.K mice. We also examined TCR Vβ repertoire of CD8^+^ T cells in B6.K HSC reconstituted gp250 SC chimeras, and the TCR Vβ repertoire was comparable to CD8^+^ T cells in B6.K HSC reconstituted B6.K chimeras (Figure 2—figure supplement 1). In marked contrast, certain Vβ usages in gp250-selected CD4^+^ T cells were decreased, including Vβ2, Vβ4, and Vβ8.1/8.2 (Figure 2—figure supplement 1). Overall, we only noticed the significant change in MHCII-restricted positive selection but not MHCI-restricted positive selection. We reasoned this observation was in line with what was expected, because we had only manipulated the class II peptides being presented. The new data are included in the newly added Figure 2 and Figure 2—figure supplement 1.

*3) Please describe in more detail the B6K>gp250 chimeras and why this relates to negative selection (i.e., increases the selecting self peptides...)*.

We agree with the reviewers that we did not state clearly the purpose of the bone marrow chimera experiments and the reasons why negative selection can be restored through this approach. We have now revised the manuscript as suggested and explained in detail in the Results section where we first mentioned the bone marrow chimera experiments.

*4)*
Table 3*: the authors might want to reference the initial sequence description of many of the TCR sequences presented, Science, March, 1988. The description and points made in*
Figure 4
*are unclear. Please re-write this section. Again, in*
Figure 4*, please make clear why this experiment restores negative selection*.

We thank the reviewers for pointing out the missed citation and unclear parts in the manuscript. We now added the mentioned reference in the main text and Table 3, and also revised the description of Figure 4.

*5) It is not entirely clear how the present data suggest a model different from that of Barton and Rudensky. Did B and R really specify that all the T cell clones specific for a given antigen-peptide originate from a single positively selecting peptide? Or is the proposed model more of a modification of the B and R proposal*?

We agree with the reviewers that the comparison between our data and previous models was not clear. We have generated a revised figure to address this concern (Figure 7). In previous work, Barton and Rudensky compared the OVA or HEL specific responses of CD4^+^ T cells selected by four different single peptide ligands, and found no difference among the T cell proliferation against OVA or HEL in these four mice. While conducting mixed lymphocyte culture with wild type antigen presenting cells, they found T cells from these four single peptide mice exhibited different degrees of proliferation against self-peptides. They concluded each single positively selecting ligand was capable of selecting CD4^+^ T cells specific for the antigens they tested, but each postselection population was unique in terms of self-reactivity. In gp250 SC mice with the well-established MCC-specific T cell features, we argued one positively selecting self-peptide enhanced the selection of certain TCRs with conserved CDR3 features to dominate in a given antigen response (such as gp250 favors the selection of MCC-specific T cells). The revised illustration model is now provided in Figure 7.

*7) Overall there are some very rough stretches of writing*.

The text of this version has been extensively revised to develop a more clear and cohesive storyline that focuses on and better explains the major findings of our study. Unclear statements, typos, and mistakes have also been corrected.